# High-Throughput Screens of Repurposing Hub and DOS Chemical Libraries Reveal Compounds with Novel and Potent Inhibitory Activity Against the Essential Non-Neuronal Acetylcholinesterase of *Schistosoma mansoni* (SmTAChE)

**DOI:** 10.3390/ijms26115415

**Published:** 2025-06-05

**Authors:** Patrick J. Skelly, Akram A. Da’dara

**Affiliations:** Molecular Helminthology Laboratory, Department of Infectious Disease and Global Health, Cummings School of Veterinary Medicine, Tufts University, North Grafton, MA 01536, USA; patrick.skelly@tufts.edu

**Keywords:** schistosomiasis, helminth, schistosome, drug therapy, enzyme inhibition

## Abstract

Schistosomiasis is a parasitic disease caused by helminth parasites of the genus *Schistosoma*, affecting >200 million people worldwide. Current schistosomiasis treatment relies on a single drug, praziquantel, highlighting the urgent need for new therapies. We have identified a non-neuronal tegumental acetylcholinesterase from *Schistosoma mansoni* (SmTAChE) as a rational and molecularly defined drug target. Molecular modeling reveals significant structural differences between SmTAChE and human AChE, suggesting the potential for identifying parasite-specific inhibitors. Here, we screened recombinant SmTAChE (rSmTAChE) against two chemical libraries: the Broad Institute Drug Repurposing Hub (5440 compounds) and the Diversity-Oriented Synthesis (DOS)-A library (3840 compounds). High-throughput screening identified 116 hits from the Repurposing Hub (2.13% hit rate) and 44 from the DOS-A (1.14% hit rate) library that inhibited rSmTAChE ≥60% at 20 µM. Dose–response assays using both rSmTAChE and recombinant human AChE (rHsAChE) revealed 19 Repurposing Hub compounds (IC_50_: 0.4–24 µM) and four DOS-A scaffolds (IC_50_: 13–29 µM), with higher selectivity for rSmTAChE. Selective inhibitors such as cepharanthine, primaquine, mesalazine, and embelin emerged as promising candidates for further evaluation in schistosomiasis treatment. These 23 newly identified selective hits provide a foundation for the further development of novel anti-schistosome therapies.

## 1. Introduction

Schistosomes are intravascular helminth parasites that are commonly known as blood flukes. They are the causative agent of schistosomiasis—a disease of major public health importance, afflicting more than 200 million people worldwide, and with over 800 million people at risk of infection [1,2,3,4,5,6]. There is no vaccine to prevent schistosome infection; control of this disease relies on the use of a single drug, praziquantel (PZQ) [7]. PZQ has now been in wide use for >40 years [8], and while it is effective against adult parasites, it does not kill juveniles [9,10]. Furthermore, reinfection following treatment is common [11,12]. Concern about the development of resistance to PZQ in the worm population has generated an urgent need to develop new therapies to treat schistosomiasis [13,14].

In this work, we aim to confront this challenge by identifying the lead compound inhibitors of a key schistosome acetylcholinesterase (AChE) enzyme. In animals, AChE (EC. 3.1.1.7) is an essential enzyme that is mostly found anchored to cell membranes at postsynaptic neuromuscular junctions, especially in nerves and muscles. Here, it can effectively terminate cholinergic transmission by rapidly hydrolyzing the neurotransmitter acetylcholine (ACh) [15]. AChE is found in both animals’ peripheral and central nervous systems. It is also found on red blood cell membranes, where it constitutes the Yt blood group antigen [16]. In humans, a single gene encodes the AChEs expressed in all tissues; alternative splicing leads to the generation of the GPI-anchored form on red blood cells.

In schistosome parasites, in addition to the expected neuronal AChE activity, AChE activity is also associated with the tegument (skin) of intravascular-stage worms [17,18,19,20]. Live adult male and female *Schistosoma mansoni*, as well as living larval stages (schistosomula), have the ability to cleave exogenous acetylthiocholine. Indeed, the majority of total adult male worm AChE activity (~70%) is attributable to the action of the surface AChE [21]. The AChE activity displayed by living worms is blocked in the presence of the selective AChE inhibitor BW284c5.1 [21,22]. Treatment of live schistosomes with Phosphatidylinositol-Specific Phospholipase C (Pi-PLC) releases SmTAChE, showing that this enzyme, like its human counterpart that is expressed in red blood cells, is glycosylphosphatidylinositol (GPI) anchored [21].

Whether a single gene encodes all schistosome AChEs was a matter of some controversy [23,24], but we recently showed that, unlike humans, *S. mansoni* has two AChE genes [21]. One, which is designated SmAChE1 (Gene ID: AF279461), encodes the enzyme expressed at neuromuscular junctions, while a second, designated SmTAChE (Gene ID: OP018961), encodes the form expressed at the tegument surface [21]. Of note, SmTAChE, but not SmAChE1, is predicted to contain a leader sequence and a consensus GPI anchoring domain, supporting the biochemical evidence that the tegumental AChE is a surface-exposed, GPI-linked protein [21].

To examine the importance of the tegumental AChE enzyme for *S. mansoni* in vivo, the SmTAChE gene was suppressed in schistosomula by RNA interference, and these parasites were used to infect mice. After 6 weeks, worm burdens were compared between mice infected with the gene-suppressed schistosomula versus mice infected with a control (non-suppressed) schistosomula. Almost no parasites were recovered from mice infected with schistosomula whose SmTAChE gene was knocked down compared to controls [21]. This result shows that the function(s) carried out by SmTAChE are essential for parasite survival. These data make SmTAChE a strong rational therapeutic target—chemicals that block SmTAChE’s vital function should mimic the RNAi effect and lead to parasite death. Therefore, finding such chemicals is the goal of the present work.

At the amino acid sequence level, SmAChE1 and SmTAChE display ~35% amino acid identity [21]. This is roughly the same as that displayed by either of the schistosome AChEs versus human AChE. Given this rather moderate level of amino acid sequence identity between the schistosome AChEs versus the human enzyme, it is likely that drugs preferentially inhibiting the schistosome enzymes can be developed, which is our aim here.

We have isolated SmTAChE homologs from the other two medically important schistosome species, *S. haematobium* and *S. japonicum* [21]. All three enzymes exhibit a high degree of sequence identity (71–85%) [21]. We hypothesize that drugs inhibiting the tegumental AChE enzyme of all three major human schistosome species can be identified, and that is our goal.

In the past, cases of human schistosomiasis have been successfully treated using the drug metrifonate, an organophosphorus compound whose active metabolite, dichlorvos (2,2-dichlorovinyl dimethyl phosphate), inhibits AChEs [13,14]. It was suggested that tegumental AChE is the target for this therapy [25]. Metrifonate is no longer used as an anti-schistosome agent, given its reduced efficacy compared to the newer drug, PZQ. Additionally, metrifonate was administered in multiple doses and was found to have high specificity for human AChE, resulting in toxicity [26,27]. Despite these disadvantages of metrifonate, its successful use in humans does highlight the potential value of identifying new, potent, but more schistosome-specific AChE blockers.

To look for such SmTAChE inhibitors, we first produced functionally active rSmTAChE in a mammalian cell expression system [28]. Biochemical characterization showed that purified rSmTAChE is a true acetylcholinesterase—it exhibits the highest turnover and catalytic efficiency for acetylthiocholine compared to chemically related substrates like butyrylthiocholine or propionylthiocholine [28]. We have additionally developed and validated a high-throughput screening protocol to look for small-molecule inhibitors of SmTAChE using the Broad Institute Validation Library [28]. Through this preliminary screening, we identified several compounds that inhibited rSmTAChE. Some of these compounds were well-known AChE inhibitors, such as physostigmine (eserine), but others were identified for the first time as AChE inhibitors [28]. Notably, one of these compounds ([2-(2-fluorophenyl)ethyl]{3-methoxy-4-[2-oxo-2-(1-piperidinyl)ethoxy] benzyl} amine hydrochloride) preferentially inhibited rSmTAChE (IC_50_ = 0.74 µM) over human AChE (IC_50_ = 151 µM). With these data as a starting point, we here undertake a more extensive screen of compounds derived from two additional Broad Institute chemical libraries in order to identify a greater array of novel parasite-specific AChE inhibitors.

First, we screened the Drug Repurposing Hub, a chemical library consisting of 5440 compounds, many of which have been approved by the United States Food and Drug Administration (USFDA); other chemicals in the library are at different stages of clinical development [29]. Hits from this library would likely decrease the time and effort in bringing drugs with novel anti-schistosome activity from the bench to the bedside [29,30,31]. In the second approach, we screened a subset of compounds from the Diversity-Oriented Synthesis library (DOS)—a novel collection of ~100,000 compounds generated by chemists at the Broad Institute, using functional group pairing and a “build/couple/pair” strategy of diversity-oriented synthesis [32,33,34,35,36,37]. This library is characterized by chemical scaffold complexity with three-dimensionality that mimics natural products. Further, because of the nature of library construction, the chemistry of each compound is easily accessible and modular, leading to tractable medicinal chemistry efforts for optimization. Furthermore, several “informer subsets” of DOS libraries exist; for example, the DOS-A library of 3840 compounds was developed based on selected chemical scaffolds [38], and it is this DOS-A library that we screen here. This approach allows the initial screening of a smaller number of compounds, and based on the identified hits, other libraries derived from their scaffolds that are available can then be screened [38]. Compounds accessed through diversity-oriented synthesis are showing promise in modulating the activities of several targets, even with some that are currently considered “undruggable” [37,39].

## 2. Results and Discussion

### 2.1. In Silico Analysis of SmTAChE Predicted Structure

The predicted 3D structure of SmTAChE, modeled at https://swissmodel.expasy.org/ (accessed on 1 January 2024), is presented in Figure 1A. The equivalent HsAChE model (model 4PQE) is shown in Figure 1B, and both structures, overlain, are depicted in Figure 1C. These data show that there is, as expected, considerable overall similarity between SmTAChE and HsAChE (model 4PQE) with a root-mean-square-distance (RMSD) value of 0.48. However, a comparative analysis of both structures using PyMol reveals that there are also substantial and distinct physical differences between the two enzymes. Of note, SmTAChE has two sizable amino acid inserts that are lacking in HsAChE: Insert #1, ^322^INVAIGKHRYDAVRKYLLPRYHKQEPF^348^ is depicted in blue in Figure 1, and insert #2, ^463^RPGLAKMPSYYYNLPLTSSPKRGYYDPDTVYIHD^496^ is depicted in red. Both inserts are found at the surface of the enzyme, with insert #2 localizing at the opening of the catalytic gorge, as depicted most clearly in Figure 1D. Long arrows in the figure point to the catalytic gorge, whose base is depicted in purple. The enzyme overlay image depicted in Figure 1C reveals the many additional structural differences (depicted in cyan in Figure 1C) between the two proteins, even beyond the highlighted inserts. Such differences portend well for the selection of inhibitors that work preferentially to block the parasite enzyme but not the human one.

A closer analysis of the 3D model shows that the catalytic triad residues (S^239^, E^401^, H^553^ in SmTAChE), located at the base of the catalytic gorge, are well conserved between SmTAChE and HsAChE, as seen in the enlarged view depicted in Figure 1E. Similarly, the oxyanionic hole residues (G^151^, G^152^, and A^240^ in SmTAChE) are highly conserved (Figure 1F). These residues play a pivotal role in acetylcholine binding by interacting with its carbonyl oxygen [40]. In other locations, however, when compared to human AChE, several amino acids in SmTAChE that are predicted to contribute to enzymatic activity vary significantly. For instance, analysis of the acyl-binding pocket shows that the second and third residues are swapped in SmTAChE: Phe^297^ in HsAChE is replaced with Val^362^ in SmTAChE, and Val^407^ in HsAChE is replaced with Phe^512^ (Figure 1G). While the residues making up the choline-binding site are conserved in both proteins (Figure 1H), the Trp^115^ residue in SmTAChE has an opposite orientation compared to its equivalent (Trp^86^) in HsAChE. Furthermore, as depicted in Figure 1I, only one out of five residues in the peripheral anionic site is conserved. These changes in the structure of SmTAChE are predicted to affect enzyme kinetics and impact the ability of different inhibitors to influence the active site of each enzyme. This analysis again suggests that parasite-specific inhibitors can be identified. Indeed, in our earlier assay development and validation work, some parasite-specific inhibitors were identified [28].

### 2.2. High-Throughput Screening of the Drug Repurposing Hub and the DOS-A Library

Both the Drug Repurposing Hub library and the Diversity-Oriented Synthesis library Set A (DOS-A), comprising a total of 9280 compounds, were screened simultaneously at a final compound concentration of 20 µM in duplicate. The percentage inhibition of enzyme activity for each compound was calculated by comparison with the activity seen in the zero-compound control wells. Figure 2A is a graphical representation of all the data where compounds inhibiting rSmAChE greater than 60% are shown in blue and are considered “active”, those that inhibit activity less than 60% are in red (“inactive”), and a small number of compounds that did not reproducibly reach the 60% cut-off are shown in green (“inconclusive”). The average Z’-Factor for the 58 plates was 0.69. Figure 2B compares the results of replicate screenings; in almost all cases, the duplicate values are very close, showing good reproducibility of the assay. An arbitrary threshold of 60% inhibition of rSmTAChE in both replicates was set for the selection of active hits (green lines in Figure 2). Using this criterion, a total of 116 hits were identified in the Drug Repurposing Hub library (Appendix A). This yields an initial hit rate of active compounds of 2.13% for this library. Some of the hit compounds are known AChE inhibitors, such as physostigmine, while others (discussed below) are identified here for the first time as AChE inhibitors. Screening the DOS-A library resulted in 44 compounds that reproducibly inhibited rSmTAChE by more than 60% (Appendix A), yielding an initial hit rate of 1.14%. From both libraries, 58 hits were considered inconclusive—one replicate showed ≥60% inhibition while the other replicate did not. These compounds were excluded from further analysis.

### 2.3. Hit Validation and Parasite Specificity

To determine the potency of the most promising compounds, we performed concentration-dependent assays for all hits that resulted in a ≥60% inhibition of rSmTAChE in both replicates. These assays were performed using the same 384-well microtiter plates as for the initial screen, except that each compound was tested at eight different concentrations (3-fold serial dilutions) ranging from 40 µM to 0.018 µM; a few compounds started at 20 µM (as indicated) due to their lower concentration in the library, which limited the maximum achievable starting dose. Furthermore, to determine the selectivity of these hits for the parasite enzyme, this assay was carried out on both rSmTAChE and recombinant human acetylcholinesterase, rHsAChE, simultaneously.

The data showing dose–responsive inhibitory activity against rSmTAChE and rHsAChE of all 116 initial hits from the Drug Repurposing Hub and the 44 hits from the DOS-A libraries are provided in Appendix A, respectively. Compound structures and inhibition curves are also provided. Overall, of the 160 compounds that were initially identified, 103 (~65%) were validated when tested in a dose-dependent manner against rSmTAChE: 78 out of the 116 compounds from the Drug Repurposing Hub (67%) (Appendix A), and 25 out of the 44 compounds from the DOS-A library (57%) (Appendix A) were confirmed. The other 57 compounds out of the 160 compounds retested did not reproducibly inhibit rSmTAChE in a concentration-dependent manner, and these were eliminated from subsequent analysis.

### 2.4. Parasite-Specific Hits from the Drug Repurposing Hub Library

Among the 78 active hits from the Drug Repurposing Hub screen that were confirmed by the dose–response experiments, nineteen compounds were more potent and selective for rSmTAChE versus rHsAChE. i.e., they have a higher selectivity index (SI) with considerably lower IC_50_ values for rSmTAChE than rHsAChE (Table 1). These compounds are listed in the order of their greater relative ability to block SmTAChE vs. HsAChE (as recorded in the “Selectivity Index (SI)” column of Table 1). For example, compound #1 has an almost 50-fold greater potency against rSmTAChE (IC_50_ = 0.42 µM) compared to rHsAChE (IC_50_ = 19.5 µM). The Selectivity Index for other compounds ranges from 21-fold (for compound #2) to 1.4-fold (for compound #16). Additionally, three compounds (compounds #17–19) are active against rSmTAChE but inactive against rHsAChE, so no SI can be determined for these. The designations for each compound assigned by the Broad Institute are listed under “Compound ID”, and other names for these chemicals are given in the “Compound Name” column. The table records how much each compound blocks rSmTAChE in the standard HTS assay (“Inhibition” column) and (in the “[Comp]” column) the compound concentration (µM) at which that level of inhibition was determined. The table also notes in the “Clinical Phase” column where each compound is currently in the commercialization pipeline; some are available as drugs (“launched”); others are in clinical trials (phases 1–3, as indicated); and some are in a “preclinical” phase. Of the 19 hit compounds, eight are launched compounds, six are in clinical trials (phase 1, 2, or 3), and five are in preclinical testing (Table 1). The IC_50_ values of the selected hits ranged from 0.4 µM (Table 1, compound #1) to 24 µM (compound #15). The profile and the information about what is currently known about each compound’s mode of action (MOA) were obtained from the Repurposing Hub database (https://repo-hub.broadinstitute.org/repurposing-app, accessed on 13 November 2024) and [29] and are noted in the far-right column of Table 1. To our knowledge, many of these compounds here have been identified for the first time as acetylcholinesterase inhibitors. Below is a description of some notable compounds.

As shown in Figure 3A, compound #1 exhibited ~50-fold greater specificity for SmTAChE (IC_50_ of ~0.4 µM) compared to rHsAChE (IC_50_ of ~20 µM). Compound #1 (4-ethylphenylamino-1,2-dimethyl-6-methylaminopyrimidinium chloride) is also known as ZD-7288 and has been reported to be a selective blocker of hyperpolarization-activated cyclic nucleotide-gated channels (HCN channels) [41,42]; it has entered phase 2 clinical trials. More recent work has shown that ZD-7288 also inhibits Na^+^ currents in dorsal root ganglion neurons [43]. Whether these effects are independent of compound #1’s acetylcholinesterase inhibitory activity, as demonstrated here, is not clear. This compound has profound effects on the human heart rate and is used at very low concentrations, which may restrict its utility for non-cardiac indications like schistosomiasis [44,45].

Compound #2, cepharanthine, has an IC_50_ for rSmTAChE (0.66 µM) that is >20-fold lower than that for rHsAChE (14 µM) (Table 1). It is a bisbenzylisoquinoline alkaloid isolated from tubers of the climbing vine *Stephania cepharantha* [46,47]. Cepharanthine is an anti-inflammatory and antineoplastic compound [47,48,49,50,51,52,53] that additionally has anti-parasitic effects [54,55,56]. It has been shown to greatly decrease levels of the malaria parasite in infected mice as well as inhibit parasite growth in vitro [57]. Given its specificity for SmTAChE vs. HsAChE and the absence of both toxicity and negative side effects [58], cepharanthine makes an especially promising lead anti-schistosomal compound.

Compound #3, cisplatin, displays about 10-fold greater selectivity for SmTAChE vs. HsAChE (Table 1). Cisplatin is a platinum-containing small molecule that is in use clinically to treat a variety of cancers [59,60,61,62]. The drug has been reported to work in part by binding to DNA and inhibiting its replication [63]. It has also been reported that cisplatin can inhibit human serum butyrylcholinesterase (BChE) [64] as well as camel retinal AChE [65]. In the latter work, it was argued that by binding to the AChE–substrate complex on the peripheral anionic site, cisplatin blocked the proper positioning of the catalytic center with its substrate. More efficient binding to the schistosome AChE in like manner is one hypothesis to explain the drug’s greater ability to inhibit this enzyme compared to HsAChE. While cisplatin has been licensed for medical use for many decades, it is known to be highly toxic, which may curtail its easy use as an anti-schistosome therapeutic [66]. Cisplatin’s reported toxicity has been hypothesized to be due to its ability to inhibit HsAChE; indeed, it was determined that high concentrations of cisplatin can inhibit human erythrocyte AChE activity [67,68,69]. On the other hand, thousands of cisplatin analogs have been developed for anticancer purposes [70,71,72]; their diverse structures and mechanisms of action suggest a potential for targeting other enzymes, such as SmTAChE. Screening these compounds could identify candidates with potent and specific anti-SmTAChE inhibitors, potentially leading to new therapeutic approaches for schistosomiasis with reduced toxicity.

Among the launched drugs tested in our screen, primaquine phosphate (#14, Table 1) was identified as a selective inhibitor of SmTAChE, with an IC_50_ of ~17 µM. Primaquine, an 8-aminoquinoline-based drug, is primarily used to treat malaria [73]. Notably, it is the only drug currently effective against the dormant liver forms of *Plasmodium vivax*, making it essential for managing relapsing malaria. Additionally, it targets late-stage gametocytes of the *Plasmodium* species [74,75,76]. The exact mechanism of primaquine action is not well known; however, it is believed that it induces oxidative stress that damages parasites’ DNA, membranes, and mitochondrial function [77]. Primaquine has been reported to inhibit human red blood cell AChE, with an IC_33_ of 38 µM and an IC_67_ of 247 µM [78]. Similarly, in our study, we found that primaquine exhibits an IC_50_ of >38 µM for rHsAChE. Whether acetylcholinesterase inhibition plays a role in primaquine’s antimalarial activity against *Plasmodium* remains unclear. Importantly, primaquine has been reported to possess moderate in vitro schistosomicidal activity against both juvenile and adult worms [79,80], although the underlying mechanism of action has not been elucidated. Based on our findings, we hypothesize that primaquine exerts its effects on schistosome parasites, at least in part, by inhibiting SmTAChE.

Another effective drug against SmTAChE is embelin (compound #16, Table 1), a naturally occurring benzoquinone derivative extracted from the fruits of *Embelia* plants [81,82]. Currently, embelin is a promising drug with diverse applications, including as an anticancer agent and for treating chronic diseases [81,83]. Embelin is a potent inhibitor of the X-linked inhibitor of apoptosis protein (XIAP) [81] and is considered safe and non-toxic [83]. Of note, embelin exhibits anthelminthic activity both in vivo and in vitro [84,85]. While the mechanism of its action against nematode parasites remains unknown, our work here suggests that inhibiting worm AChE offers one explanation. Recent studies evaluating the effect of embelin on Alzheimer’s disease have shown that it can inhibit human AChE activity in vitro [86,87,88,89]. While the IC_50_ value of embelin is just slightly lower for SmTAChE (1.25 µM) compared to HsAChE (1.80 µM), given its broad pharmacological properties and safety profile, we propose further investigation of embelin as a potential anti-schistosome drug.

Compound #4 (CGP 71683), like cisplatin above, displays about 10-fold greater selectivity for SmTAChE vs. HsAChE. This compound has been described as a neuropeptide Y5 (NYP Y5) receptor antagonist [90] and has been tested as an appetite suppressant [91]. The drug (as part of the 400-chemical collection known as the “Medicines for Malaria Venture Stasis Box”) has also been tested for its impact, over 3 days, on the survival of newly transformed *S. mansoni* schistosomula as well as on adult male and female worms [92]. An IC_50_ value of ~1 µM was reported against schistosomula and of ~2 µM against adults. The drug was next tested for any impact on worms in vivo. It was administered one time (200 mg/kg) by oral gavage to four mice, but no differences in worm burden were seen in this group compared to the controls 16–18 days post-drug administration. Our data suggest that further testing of this compound over a wider range of doses is warranted.

Compound #5, liothyronine, is a therapeutic formulation of the primary physiologically active form of endogenous thyroid hormone. Compound #6, GW4064, is a synthetic isoxazole that is reported to act as an agonist of farnesoid X receptors (FXRs). Compound #7, triclosan, is widely used as an antimicrobial agent in personal care products like skin creams and soaps. All three are shown here as AChE blockers and with a similar (6- to 7-fold) increased inhibition of SmAChE vs. HsAChE in our assays. While these hits could be further developed, not all AChE inhibitors identified here would make top-tier lead anti-schistosome agents. For instance, compound #9, sodium nitroprusside, is a potent vasodilator with a half-life in the circulation of just 2 min and has high toxicity [93].

A few compounds (#17, #18, and #19) show specific inhibition of SmATChE with no inhibition detected against HsAChE under the experimental conditions used here (Table 1). For example, mesalazine (mesalamine, compound #19), a cyclooxygenase lipoxygenase inhibitor, as well as an inhibitor of nuclear factor κB activation [94,95], effectively inhibits rSmTAChE, with an IC_50_ of approximately 1.7 µM (Table 1; Figure 3B). However, mesalazine has no detectable inhibitory effect on rHsAChE. Mesalazine is primarily used as an anti-inflammatory drug for the treatment of inflammatory bowel disease (IBD), as well as ulcerative colitis (UC) and Crohn’s disease [96,97]. Mesalazine is considered a safe drug [98], and due to its effect on SmTAChE, it emerges as a new promising candidate for schistosomiasis therapy. Similarly, SR-33805 (compound #18) shows specific inhibition of SmTAChE, with no activity detected against HsAChE (Table 1). SR-33805 is a potent Ca^2+^ channel antagonist [99,100]. These compounds merit renewed scrutiny at higher concentrations in order to identify those that display the greatest specificity for the schistosome enzyme.

Several other compounds have been identified with higher specificity for SmTAChE than HsAChE, such as compounds #5 and #9 (Table 1). However, due to their mode of action and the low doses used in humans, these compounds would not presently be appropriate for schistosomiasis therapy. Nonetheless, these compounds could serve as a basis for identifying and developing other AChE inhibitors.

### 2.5. Parasite-Specific Hits from the DOS-A Library

Our screen of the DOS-A library resulted in the initial identification of 44 hit compounds that were able to inhibit SmTAChE by ≥60% at 20 µM. However, the dose–response analysis confirmed the ability of only 25 of these compounds to be able to reproducibly inhibit the enzyme (Appendix A). Among the 25 confirmed hits, just four compounds (DOS#1-4) were able to inhibit rSmTAChE with IC_50_ values lower than those for HsAChE (Table 2, Figure 4). Interestingly, a diastereomeric pair of compounds—DOS#1 (BRD0282; (2R,3R,4S)-4-(hydroxymethyl)-3-[4-[2-(3-methoxyphenyl)ethynyl]phenyl]-1-[oxo(2-pyridinyl)methyl]-2-azetidinecarbonitrile) and DOS#3 (BRD6110; (2S,3R,4R)-4-(hydroxymethyl)-3-[4-[2-(3-methoxyphenyl)ethynyl] phenyl]-1-[oxo(2-pyridinyl)methyl]-2-azetidinecarbonitrile) (Table 2)—are both inhibitory, suggesting that the core of this scaffold can serve to identify or guide the synthesis of additional analogs for SmTAChE inhibition. Altering the stereochemistry at both the hydroxymethyl and the cyano groups impacts the IC_50_ for rSmTAChE; DOS #1’s IC_50_ is 13 µM, but DOS #3’s IC_50_ is more than double at 29.4 µM. At the same time, DOS#1 and DOS#3 have very similar IC_50_ values when tested against HsAChE, suggesting that this (DOS #1/DOS #3) chemical scaffold can be used to exploit the differences in the enzymes’ active sites to improve selectivity (Figure 4A,C). DOS#2 was also more selective for SmTAChE (IC_50_ = 15 µM) than HsAChE (IC_50_ = 35.5 µM) (Figure 4B), and one additional compound (DOS#4) was effective against SmTAChE (IC_50_ = 16 µM) but was determined to be inactive against HsAChE (Table 2, Figure 4D). Furthermore, an additional 15 hits from the DOS-A library were able to weakly inhibit SmTAChE (by <60%) but were inactive against HsAChE and warrant further examination (Appendix A).

### 2.6. HsAChE-Specific Hits from the Drug Repurposing Hub and DOS-A Libraries

As described in Methods, all 160 hits that inhibited rSmTAChE by ≥60% in our library screens were also evaluated against rHsAChE in a dose–response experiment (Appendix A). Of the 116 hits from the Repurposing library that inhibited rSmTAChE, 74 were able to inhibit rHsAChE in a dose–response manner with varying IC_50_ values; the remaining 42 compounds were inactive against the human enzyme (Appendix A). Twenty-one compounds from the Drug Repurposing Hub library were notably more effective against rHsAChE than rSmTAChE (Table 3). The chemicals are listed in order of their greater selectivity for the human enzyme compared to the schistosome one (SI, selectivity index “Sm/Hs” column). The top two hits show strikingly higher specificity for HsAChE vs. SmTAChE, with demecarium bromide (#1 on the list) and Huperzine A (#2) having 531- and 147-fold greater potency for HsAChE versus SmTAChE, respectively (Table 3, Figure 3C,D). The “Clinical Phase” of each of these compounds is also noted in the table. Finally, the mode of action (MOA) of each compound, as derived from https://repo-hub.broadinstitute.org/repurposing-app (accessed on 13 November 2024), is also recorded. The best hits are known acetylcholinesterase blockers like physostigmine (compounds #3, #6, and #8 in Table 3) and its analog neostigmine (compound #4), but several quite potent HsAChE inhibitors are identified here for the first time. These data may suggest additional uses for these drugs in the treatment of other human diseases.

Regarding the DOS-A library, 36 out of 44 compounds were determined to be inactive against rHsAChE. The remaining eight compounds were determined to be weak inhibitors of rHsAChE, with IC_50_ values ranging from 31–38 µM (Appendix A).

## 3. Materials and Methods:

### 3.1. Expression and Purification of rSmTAChE

A Chinese Hamster Ovary Cell line (CHO-S), stably expressing a secreted form of recombinant SmTAChE (rSmTAChE), was previously generated [28]. Stable cell lines were grown in serum-free Freestyle Expression Medium supplemented with 8 mM L-glutamine (ThermoFisher Scientific, Waltham, MA, USA) at 37 °C and 8% CO_2_ in vented culture flasks with shaking at 140 rpm, as previously described [28]. rSmTAChE was purified from the cell culture medium by standard Immobilized Metal Affinity Chromatography (IMAC) using HisTrap™ Excel columns as described by the manufacturer (GE Healthcare Life Sciences, Marlborough, MA, USA). The purified recombinant protein was dialyzed overnight at 4 °C against phosphate-buffered saline (PBS), then concentrated by ultrafiltration centrifugation using Pierce Protein Concentrators (10K MWCO, ThermoFisher Scientific). Protein purity was assessed using SDS-PAGE, and protein concentration was determined using a BCA Protein Assay Kit (Pierce, Rockford, IL, USA).

The 3D structure of SmTAChE was modeled using the Swiss-Model online tool (https://swissmodel.expasy.org/, accessed on 1 January 2024) [101]. Analysis of the predicted structure was performed using PyMol V2.5.7 (Schrodinger, LLC. Cambridge, MA, USA).

### 3.2. SmTAChE Activity

AChE activity was measured at room temperature (~25 °C) by the modified Ellman method using acetylthiocholine iodide (ATCh; Sigma-Aldrich, St. Louis, MO, USA) as substrate [21,28,102]. The standard reaction mixture (200 µL) contained 1 mM acetylthiocholine iodide (ATCh) and 1 mM 5,5′-dithiobis (2-nitrobenzoic acid) (DTNB) in 100 mM sodium phosphate (pH 7.2). Absorbance at 412 nm was monitored over 1 h using a Synergy HT spectrophotometer (Bio-Tek Instruments, Winooski, VT, USA).

### 3.3. Chemical Libraries

Two chemical libraries were tested: the first library is the Broad Repurposing Hub library (Broad Institute, Cambridge, MA, USA), which consists of 5440 different compounds with 663 different therapeutic indications and over 2000 different targets [29]. Drugs in this collection are approved by the US FDA or have undergone testing in at least one phase of a clinical or preclinical trial [29]. Because of this, pharmacodynamics, pharmacokinetics, safety, and toxicity for humans have been characterized for most of these compounds. The second library is the Diversity-Oriented Synthesis library set A (DOS-A), containing 3840 screening compounds with defined chemical scaffolds [33,34].

### 3.4. Screening of the Chemical Libraries for SmTAChE Inhibitors

To prepare for conducting a high-throughput screen (HTS) of potential SmTAChE inhibitors, the AChE assay based on Ellman’s reagent, as described above, was first modified to a 384-well plate format as described [28].

All screening procedures were performed at the Center for the Development of Therapeutics (CDoT) at the Broad Institute of MIT and Harvard using our previously validated assay in a 384-well plate format [28]. Briefly, compounds in the Repurposing or DOS-A libraries were pre-plated in individual wells of flat-bottom microtiter plates (Corning Incorporated, Corning, NY, USA). Each well contained nanoliter volumes of one individual compound, calculated in a manner such that the addition of the enzyme and substrate would result in a final screening concentration of 20 µM; however, a few compounds were tested at 5 µM or 10 µM due to their low concentrations in the library (as indicated). A total of 58 plates were used and bar-coded for identification. Wells in columns 2 through 22 contained individual pre-spotted compounds, while columns 1 and 24 contained an equivalent volume of DMSO (as a negative—no chemical inhibition—control), and selected wells in column 23 contained physostigmine (a known AChE inhibitor, and at a final concentration of 20 µM) as a positive control for AChE blockage. On the day of screening, the plates were allowed to warm up to room temperature, and 30 µL of rSmTAChE (5 ng/well), diluted in 100 mM sodium phosphate buffer (pH 7.2), was dispensed in individual wells in columns 2 through 24, while buffer alone was dispensed into column 1’s (negative control/blank) wells. The plates were then incubated at room temperature for 20 min. Then, 20 µL of the substrate solution was added to all wells (the final substrate concentration was 0.35 mM acetylthiocholine and 0.35 mM Ellman reagent (DTNB) in 100 mM sodium phosphate buffer, pH 7.2). This substrate concentration was chosen because it is roughly equivalent to the K_m_ value of rSmTAChE [28]. Assay plates were read twice at 405 nm (using a Perkin Elmer Envision multimode plate reader)—immediately after the addition of substrate and again 20 min later. Screening data were analyzed using Genedata Assay Analyzer 10.0.2 Standard. The data were normalized to compound wells. SciTegic Pipeline Pilot 7.0 was used to pair duplicate data points between the two validation runs prior to the creation of the screening graphics with TIBCO Spotfire 3.3.1. Compounds resulting in ≥60% inhibition in rSmAChE activity in 2 independent replicates were selected for further evaluation, as described below. The identification of the compounds, their structure, MOA, and other features was examined using the Broad Institute Drug Repurposing Hub data portal, https://repo-hub.broadinstitute.org/repurposing-app (accessed on 13 November 2024) [29].

### 3.5. Hit Validation Using Concentration-Dependent Assays (IC_50_) and Parasite Specificity Determination

All compounds that resulted in ≥60% inhibition in both replicate screens were selected for validation analysis in concentration-dependent assays. The IC_50_ values of the selected compounds, including the positive controls, were determined as described previously [28]. Briefly, the test compounds prepared in DMSO were serially diluted (3-fold for 8 different dilutions). The starting concentration was 40 µM. To identify the schistosome-specific inhibitors, both rSmTAChE and commercially obtained recombinant human AChE (rHsAChE; Sigma) were tested side-by-side. The amount of enzyme used in each assay was determined to be in the linear range of activity. Enzymes were incubated with test compounds for 20 min, and then the reaction was started by the addition of substrate solution (0.35 mM ATCh and 0.35 mM DTNB), as above. The rate of the reactions was determined, and the residual enzyme activity (percentage of control) was calculated and plotted against the log inhibitor concentration (Log_10_[I]). All data were created and analyzed by Genedata Screener. Additional IC_50_ graphs were generated using nonlinear regression analysis of a log[I] vs. normalized response-variable slope using GraphPad Prism v. 10.4. The Selectivity Index (SI) was calculated as the ratio of the IC_50_ for the human enzyme to the IC_50_ for the schistosome enzyme; a higher SI value indicates greater selectivity for the schistosome enzyme. Conversely, to determine the SI for the human enzyme, the IC_50_ for the schistosome enzyme was divided by the IC_50_ for the human enzyme.

### 3.6. Statistical Analysis

The student’s *t*-test and one-way analysis of variance (ANOVA) with Tukey’s post hoc analysis were used to compare the means between a target group and a control group, and *p*-values less than 0.05 were considered significant. Data were assessed for normality using Shapiro–Wilk tests with GraphPad Prism 10.4. Bartlett’s test for homogeneity of variances was used to confirm the assumption of equal variances.

## 4. Conclusions

We have identified and characterized a molecularly defined target to treat schistosomiasis, namely the tegumental AChE enzyme, SmTAChE. RNAi experiments have revealed the vital nature of the enzyme for the parasites, so drugs that mimic this effect should kill the worms. We have produced and validated a high-throughput screening strategy in order to identify novel SmTAChE inhibitors. Our strategy of screening the schistosome enzyme alongside its human counterpart led to the identification of schistosome-specific inhibitors and is designed to minimize adverse side effects. Screening of the Repurposing Hub library, as described here, has successfully identified several selective SmTAChE inhibitors, with cepharanthine, primaquine, mesalazine, and embelin emerging as especially promising candidates for further evaluation in schistosomiasis treatment. Furthermore, screening of the DOS-A library has identified four different scaffolds that can be used as a basis to develop more schistosome-specific AChE inhibitors. Future work will focus on in vivo testing, structural optimization, and an assessment of pharmacokinetic properties to advance these compounds toward clinical deployment.

## Figures and Tables

**Figure 1 ijms-26-05415-f001:**
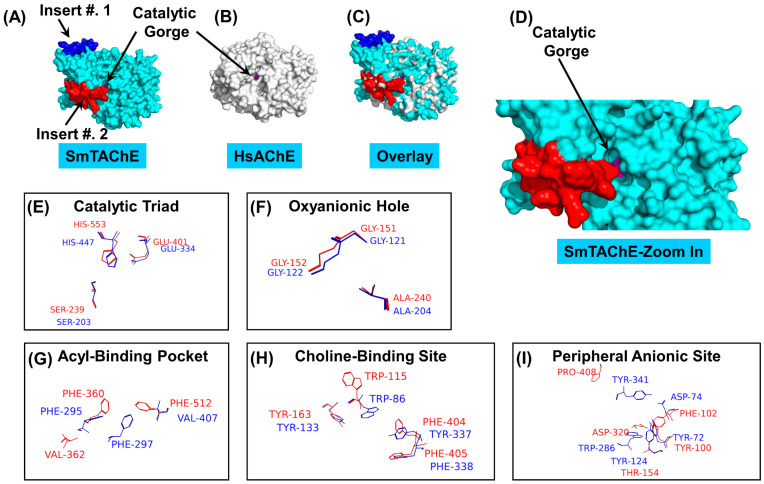
A 3D structural model of SmTAChE. Surface representations of the homology models of SmTAChE (**A**) created by the Swiss-Model server and HsAChE (4PQE model) (**B**). Both proteins are depicted superimposed in (**C**). The enzymes’ catalytic gorges are indicated by long arrows, and the base of each gorge is indicated in purple. Sizable differences between the two proteins are depicted as SmTAChE insert #1 (blue) and insert #2 (red), with the remainder of SmTAChE in cyan. A zoomed-in view of the active gorge of SmTAChE, depicting the proximity of insert #2 (red) in relation to the opening of the catalytic gorge, is seen in (**D**). Depiction of the relative arrangement of the catalytic triad residues (**E**) and the residues forming the oxyanionic hole (**F**) of SmTAChE (red) versus HsAChE (blue). (**G**) Depiction of residues in the acyl-binding pocket in the schistosome and human enzymes shows that Phe297 is substituted with Val362 in SmTAChE. Residues comprising the choline-binding site of both proteins are depicted in (**H**); while most residues are conserved, SmTAChE Trp115 exhibits a different orientation compared to HsAChE Trp86. The peripheral anionic site of SmTAChE, compared to HsAChE (shown in (**I**)), contains only one conserved residue (Tyr100) out of five residues that constitute the site. In all cases, the SmTAChE residues are shown in red, and the HsAChE residues are in blue. Images were created in PyMol software.

**Figure 2 ijms-26-05415-f002:**
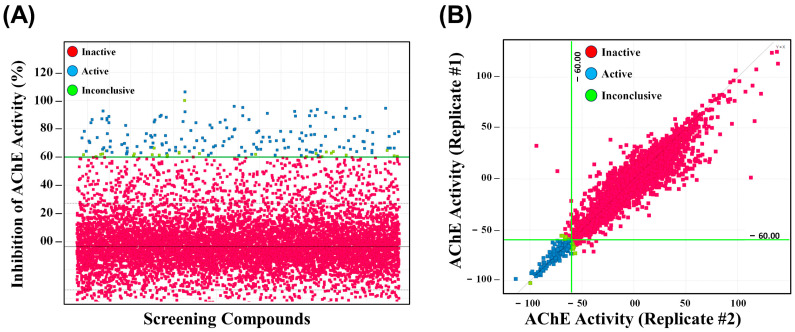
Chemical screen performance and reproducibility. (**A**) A graphical representation of the results from high-throughput chemical screens for inhibitors of rSmTAChE. The red symbols represent compounds considered inactive, the blue symbols represent active compounds (≥60% inhibition), and the green symbols represent inconclusive hits. The horizontal green line indicates the 60% inhibition, an arbitrary threshold for initial hit identification. (**B**) Recombinant SmTAChE activity was monitored in the presence of test compounds, and data are presented relative to the 100% activity levels measured in the absence of any chemical. The red symbols represent inactive compounds, the blue symbols represent active compounds (≥60% inhibition), and the green symbols represent inconclusive hits. The horizontal and vertical green lines indicate the 60% arbitrary threshold for initial hit identification. The two replicates are plotted against each other and show a linear fit of the dataset, indicating good reproducibility.

**Figure 3 ijms-26-05415-f003:**
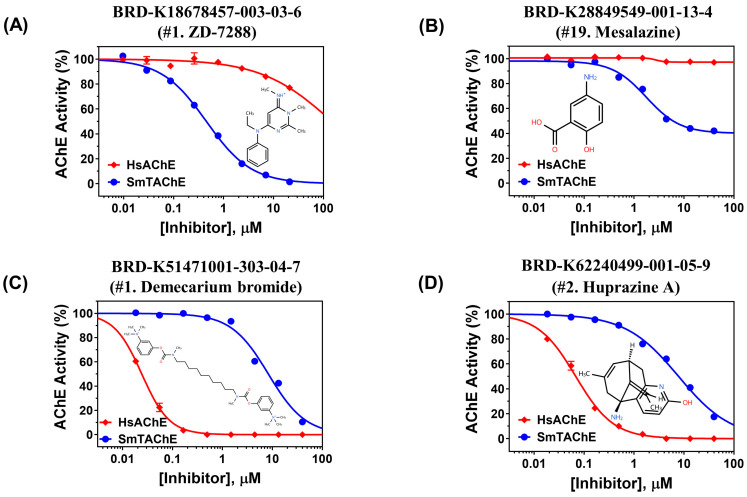
Dose–response graphs of selected test compounds from the Drug Repurposing Hub. (**A**,**B**) shows data from two compounds that exhibit higher specificity for rSmTAChE vs. HsAChE (compounds #1 and #19 in Table 1). In contrast, (**C**,**D**) show data from two compounds that exhibit higher specificity for HsAChE vs. rSmTAChE. The red lines represent the human enzyme responses, while the blue lines represent the schistosome enzyme responses, as indicated. The graphs were generated using GraphPad Prism (V. 10.4). Chemical structures of the tested compounds are depicted.

**Figure 4 ijms-26-05415-f004:**
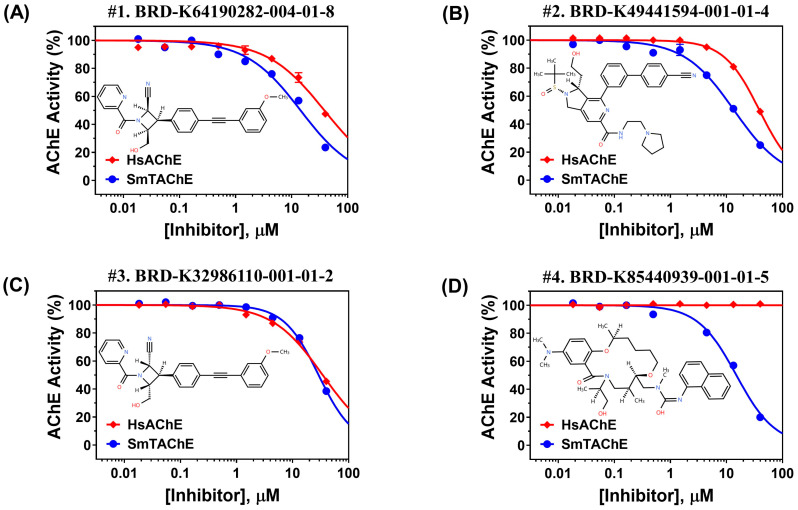
Dose–response graphs of the test compounds identified from the DOS-A library. Activity of the selected DOS-A compounds (**A**–**D** represent DOS #1–4 in Table 2) against rSmTAChE (blue lines) and rHsAChE (red lines). The graphs were generated using GraphPad Prism (V. 10.4). Chemical structures of the tested compounds are depicted.

**Table 1 ijms-26-05415-t001:** rSmTAChE-specific hits from the Drug Repurposing Library.

No.	Compound ID	Compound Name	HTS/rSmTAChE		IC_50_ (µM)		SI ^$^ (Hs/Sm)	Clinical Phase	MOA
			[Comp.] (µM)	Inhibition (Ave, %)	rSmTAChE	rHsAChE			
1	BRD-K18678457-003-03-6	ZD-7288	10	75.00	0.416	19.5	46.87	Phase 2	HCN channel blocker
2	BRD-K96194081-001-11-0	Cepharanthine	10	66.29	0.655	13.9	21.22	Phase 2	NFκB pathway inhibitor
3	BRD-K69172251-001-08-9	Cisplatin	20	89.23	3.81	38.0	9.97	Launched	DNA alkylating agent, DNA synthesis inhibitor
4	BRD-K34321528-003-02-0	CGP-71683	5	61.37	2.21	19.5	8.82	Preclinical	Neuropeptide receptor antagonist
5	BRD-K89152108-236-06-8	Liothyronine	20	78.78	5.21	38.0	7.29	Launched	Thyroid hormone stimulant
6	BRD-K88186167-001-04-8	GW-4064	20	66.00	1.47	8.83	6.00	Preclinical	FXR agonist
7	BRD-K41731458-001-15-1	Triclosan	20	68.01	6.57	38.0	5.78	Launched	Antibacterial agent
8	BRD-K14991967-001-02-6	GSK-650394	10	75.44	2.34	8.1	3.46	Preclinical	Serum glucocorticoid-regulated kinase inhibitor
9	BRD-K48526231-304-03-6	Sodium Nitroprusside	20	86.41	10.02	26.02	2.60	Launched	Nitric oxide donor
10	BRD-K39841531-001-02-1	TG-101209	20	71.00	6.13	15.4	2.51	Preclinical	JAK inhibitor
11	BRD-A87130939-001-07-9	Masoprocol	20	76.44	16.4	38.0	2.32	Launched	Lipoxygenase inhibitor
12	BRD-K58501140-002-01-0	TAK-875	20	65.04	16.9	38.0	2.25	Phase 3	Insulin secretagogue
13	BRD-K45906612-001-01-8	Presatovir	20	66.88	17.2	38.0	2.21	Phase 2	RSV fusion inhibitor
14	BRD-A55913614-316-09-6	Primaquine phosphate	20	73.71	17.2	38.0	2.21	Launched	DNA inhibitor; antimalarial agent
15	BRD-K22149900-001-05-4	Ceritinib	20	66.98	24.0	38.0	1.58	Launched	ALK Tyrosine Kinase Receptor Inhibitor
16	BRD-K86727142-001-12-4	Embelin	10	88.65	1.25	1.80	1.44	Preclinical	HCV inhibitor, XIAP inhibitor
17	BRD-K16732600-001-01-7	MK-0893	20	88.71	3.11	Inactive *	ND ^#^	Phase 2	Glucagon Receptor antagonist
18	BRD-K43002771-034-02-4	SR-33805	5	74.26	6.82	Inactive	ND	Phase 1	Calcium channel blocker
19	BRD-K28849549-001-13-4	Mesalazine	10	66.7	1.70	Inactive	ND	Launched	Cyclooxygenase inhibitor, lipoxygenase inhibitor

^$^ SI: The Selectivity Index (SI) was calculated by dividing the IC_50_ for HsAChE by the IC_50_ for SmTAChE. * Inactive at the maximum tested concentration (40 µM) in the dose–response analysis. ^#^ ND: value cannot be determined.

**Table 2 ijms-26-05415-t002:** rSmTAChE-specific hits from the DOS-A library.

No.	Compound ID (Name)	rSmTAChE HTS		IC_50_ (µM)		SI ^$^ (Hs/Sm)
		[Comp.] (µM)	Inhibition (Ave, %)	rSmTAChE	rHsAChE	
1	BRD-K64190282-004-01-8 (BRD0282)	20	92.21	13.0	33.7	2.59
2	BRD-K49441594-001-01-4	20	61.81	15.1	35.5	2.35
3	BRD-K32986110-001-01-2 (BRD6110)	20	71.41	29.4	31.2	1.1
4	BRD-K85440939-001-01-5	20	63.89	16.2	Inactive *	ND ^#^

^$^ SI: The Selectivity Index (SI) was calculated by dividing the IC_50_ for HsAChE by the IC_50_ for SmTAChE. * Inactive at the maximum tested concentration (40 µM) in the dose–response analysis. ^#^ ND: value cannot be determined.

**Table 3 ijms-26-05415-t003:** rHsAChE-specific hits from the Drug Repurposing Library.

No.	Compound ID	Compound Name	rSmTAChE HTS		IC_50_ (µM)		SI ^$^ (Sm/Hs)	Clinical Phase	MOA
			[Comp.] (µM)	Inhibition (Ave, %)	rSmTAChE	rHsAChE			
1	BRD-K51471001-303-04-7	Demecarium bromide	20	65.33	11.93	0.02	531.20	Launched	Acetylcholinesterase inhibitor
2	BRD-K62240499-001-05-9	Huperzine A	20	64.07	8.81	0.06	146.97	Phase 2	Acetylcholinesterase inhibitor
3	BRD-K69688083-004-23-1	Pyridostigmine bromide	20	92.44	0.87	0.018	48.41	Launched	Acetylcholinesterase inhibitor
4	BRD-K18922609-004-23-1	Neostigmine bromide	20	88.11	3.79	0.16	24.28	Launched	Acetylcholinesterase inhibitor
5	BRD-K12068470-001-02-5	LY2608204	20	76.45	12.00	0.52	23.28	Phase 2	Glucokinase activator
6	BRD-K25650355-065-02-0	Physostigmine sulfate	20	100.00	0.47	0.08	5.92	Launched	Acetylcholinesterase inhibitor
7	BRD-K72029282-001-22-0	Probucol	20	93.68	38.00	7.09	5.36	Launched	Atherogenesis inhibitor
8	BRD-K25650355-059-19-7	Physostigmine salicylate	20	91.99	0.77	0.15	5.10	Launched	Acetylcholinesterase inhibitor
9	BRD-K87700323-003-05-1	Cetylpyridinium chloride	20	71.00	7.37	1.47	5.01	Launched	Gingivitis- antiseptic
10	BRD-K29656036-001-02-5	MK-8245	20	71.49	37.35	7.93	4.71	Phase 2	Stearoyl-CoA desaturase inhibitor
11	BRD-A71774530-001-05-9	Lufenuron	20	92.64	38.00	12.00	3.17	Launched	Chitin inhibitor
12	BRD-K51899933-001-02-6	Azeliragon	20	78.5	15.55	5.00	3.11	Phase 3	RAGE receptor antagonist
13	BRD-K13387373-004-14-5	Thonzonium bromide	20	66.98	1.12	0.38	2.92	Launched	ATPase inhibitor
14	BRD-K29415052-050-05-5	NVP-BGT226	10	84.34	38.00	13.5	2.81	Phase 1/2	PI3K Inhibitor
15	BRD-M30288325-001-01-4	G15	20	77.99	38.00	13.5	2.81	Preclinical	Estrogen receptor antagonist
16	BRD-K95523387-001-09-6	OLDA	20	94.46	36.07	13.5	2.67	Preclinical	TRPV agonist
17	BRD-K97045029-001-04-3	Pranlukast	20	66.30	27.34	12.57	2.18	Launched	Leukotriene receptor antagonist
18	BRD-K98251413-001-06-5	IOX2	5	66.25	16.73	7.75	2.16	Preclinical	Hypoxia-inducible factor inhibitor
19	BRD-K35367061-001-01-1	LY223982	20	80.52	15.18	7.22	2.1	Phase 2	Leukotriene receptor antagonist
20	BRD-K84544951-236-01-0	Sodium-tetradecyl-sulfate	20	74.65	11.38	5.66	2.01	Launched	Sclerosing agent
21	BRD-K22127577-001-03-7	Crenolanib	20	64.48	19.5	9.91	1.97	Phase 2	PDGFR tyrosine kinase receptor inhibitor

^$^ SI: The Selectivity Index (SI) was calculated by dividing the IC_50_ for SmTAChE by the IC_50_ for HsAChE.

## Data Availability

The data generated during this study are included in the published manuscript. The raw data supporting the conclusions of this article will be made available by the authors upon request.

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
