# Peer review of "High-Throughput Screens of Repurposing Hub and DOS Chemical Libraries Reveal Compounds with Novel and Potent Inhibitory Activity Against the Essential Non-Neuronal Acetylcholinesterase of Schistosoma mansoni (SmTAChE)"

_ijms, 2025, doi:10.3390/ijms26115415_

Round 1
Reviewer 1 Report
Comments and Suggestions for Authors
The present manuscript by Skelly et al. concerns the discovery of non-neuronal acetylcholinesterase inhibitor compounds of Schistosoma mansoni, with the aim of developing new therapies for schistosomiasis. An investigation was conducted into two significant chemical libraries aimed at identifying specific inhibitors for SmTAChE. In silico structural analysis revealed significant similarities and differences between SmTAChE and human AChE, suggesting opportunities for the development of parasite-specific inhibitors.
In order to establish the potency and specificity of the inhibitors that had been identified, validation experiments were conducted, focusing on the determination of the activity against rSmTAChE in comparison to rHsAChE.
The work is both well-conceived and well described in the paper.
It is imperative to note the following observation regarding the introduction: Information contained within the introduction from lines 111 to 128 should be incorporated into the subsequent paragraph.
It is evident that the text would benefit from a thorough revision to address the grammatical errors and the use of scientific language. The utilisation of numerical values to denote compounds instead of the conventional alphabetic coding is a notable observation.
In light of the aforementioned remarks, it is evident that the paper requires minor revisions prior to its final acceptance for publication.
Author Response
The present manuscript by Skelly et al. concerns the discovery of non-neuronal acetylcholinesterase inhibitor compounds of Schistosoma mansoni, with the aim of developing new therapies for schistosomiasis. An investigation was conducted into two significant chemical libraries aimed at identifying specific inhibitors for SmTAChE. In silico structural analysis revealed significant similarities and differences between SmTAChE and human AChE, suggesting opportunities for the development of parasite-specific inhibitors.
In order to establish the potency and specificity of the inhibitors that had been identified, validation experiments were conducted, focusing on the determination of the activity against rSmTAChE in comparison to rHsAChE.
The work is both well-conceived and well described in the paper.
Comment #1. It is imperative to note the following observation regarding the introduction: Information contained within the introduction from lines 111 to 128 should be incorporated into the subsequent paragraph.
Many thanks to the reviewer for the positive feedback.
Response: This paragraph introduces the two libraries used in the screens while the results of these screens are presented in subsequent sections. We feel that this is a logical progression and, unless the editor feels otherwise, our preference is to retain the text it in its current position.
Comment #2. It is evident that the text would benefit from a thorough revision to address the grammatical errors and the use of scientific language. The utilisation of numerical values to denote compounds instead of the conventional alphabetic coding is a notable observation.
Response: We carefully reviewed the manuscript to ensure it is free of grammatical errors and consistently uses correct scientific language to describe the methods and results. Unless the reviewer highlights specific issues, we do not see language problems in the manuscript (nor did the other three reviewers).
Regarding the issue of numerical values, for the drug repurposing compounds, we provide both the compound numbers and their common names. For the four DOS compounds, we also list the numbers along with their chemical names. The intent is to offer readers a clear reference to the compounds being discussed, rather than to emphasize the naming.
Reviewer 2 Report
Comments and Suggestions for Authors
The manuscript entitled “ High-Throughput Screens of Repurposing Hub and DOS Chemical Libraries reveal Compounds with Novel and Potent Inhibitory Activity against the Essential Non-Neuronal Acetyl-cholinesterase of Schistosoma mansoni (SmTAChE) ” by Skelly and Da’dara is an interesting study, and it should be published.
All the sections are well written, and the literature review is exhaustive. The only obvious pitfall is the relatively low quality of structural formulas in Figures 3C, 3D, and 4A-D. I do understand that maybe this is only due to the lower resolution of figures in the review version of the text. But please double-check and improve if needed.
Author Response
Comment 1. All the sections are well written, and the literature review is exhaustive. The only obvious pitfall is the relatively low quality of structural formulas in Figures 3C, 3D, and 4A-D. I do understand that maybe this is only due to the lower resolution of figures in the review version of the text. But please double-check and improve if needed.
Many thanks to the reviewer for the positive feedback.
Response: Thank you for pointing this out. The reviewer is correct—the PDF review version displays the structures at lower resolution. However, we did submit the figures in high resolution, and we will work with the editor to ensure that the final version includes the high-quality images, as intended.
Reviewer 3 Report
Comments and Suggestions for Authors
The authors screened for anti-parasite-drugs using Schistosoma mansoni acetylcholinesterase (SmTAChE) as a drug target. The experimental results are clear and the materials and methods are described in detail. I only have below small suggestions:
Line 13: Authors should provide the full name of Schistosoma mansoni when it is first mentioned in the article.
Line 60-62: The authors should provide gene IDs of SmAChE1 and SmTAChE.
Line221-222:The author should explain why some compounds were started at 20 µM.
Author Response
The authors screened for anti-parasite-drugs using Schistosoma mansoni acetylcholinesterase (SmTAChE) as a drug target. The experimental results are clear and the materials and methods are described in detail. I only have below small suggestions:
Many thanks to the reviewer for their time and valuable comments.
Comment 1. Line 13: Authors should provide the full name of Schistosoma mansoni when it is first mentioned in the article.
Response: We have done that in the revised manuscript (line 13).
Comment 2. Line 60-62: The authors should provide gene IDs of SmAChE1 and SmTAChE.
Response: As requested, we added the gene ID numbers in the revised manuscript (lines 60 to 62).
Comment 3. Line221-222:The author should explain why some compounds were started at 20 µM.
Response: The reason some compounds were started at 20 µM was due to their lower concentrations in the library, which limited the maximum achievable starting dose and we have made this clear in the revised manuscript (lines 220-221).
Reviewer 4 Report
Comments and Suggestions for Authors
Dear authors,
Congratulations for the MS “High-Throughput Screens of Repurposing Hub and DOS 2 Chemical Libraries reveal Compounds with Novel and Potent Inhibitory Activity against the Essential Non-Neuronal Acetylcholinesterase of Schistosoma mansoni (SmTAChE)” submitted to IJMS. The MS is well-written and comprises a relevant study in the IJMS scopes. However, I believe there is so much repetitive text, and so recommend minor revisions to improve the main text. Please, find below my suggestions:
- The keywords should be words not presented in the title. It´s make to facilitate the search for similar articles with words didn´t find in the title;
- The paragraph between lines 86 and 94 is unnecessary. It seems more like a discussion than something extremely required to understand the research;
- The sentences in lines 165-166 and 222-223 are not necessary since these statements were exhaustively repeated in other parts of the main text;
- Figure 1 should be presented before 2.2 sub-section to improve reading;
- The concentration of the calculated IC50 could be in micromolar in the tables, as represented in the main text.
- The “ratio” mentioned in the main text and in the tables is at really the selective index (SI) present in others chemotherapy studies. Please, consider adjusting to SI your results.
Best regards.
Author Response
Congratulations for the MS “High-Throughput Screens of Repurposing Hub and DOS 2 Chemical Libraries reveal Compounds with Novel and Potent Inhibitory Activity against the Essential Non-Neuronal Acetylcholinesterase of Schistosoma mansoni (SmTAChE)” submitted to IJMS. The MS is well-written and comprises a relevant study in the IJMS scopes. However, I believe there is so much repetitive text, and so recommend minor revisions to improve the main text. Please, find below my suggestions:
Comment 1. The keywords should be words not presented in the title. It´s make to facilitate the search for similar articles with words didn´t find in the title;
First, we thank the reviewer for the kind congratulations and the positive feedback.
Response. We thank the reviewer for the suggestion. In response, we removed four keywords and added two new ones to the list (line 26).
Comment 2. The paragraph between lines 86 and 94 is unnecessary. It seems more like a discussion than something extremely required to understand the research;
Response: We understand the reviewer's perspective; however, we believe that this paragraph further strengthens the rationale for targeting AChE as a drug target. Therefore, if the editor agrees, we would prefer to retain it.
Comment 3. The sentences in lines 165-166 and 222-223 are not necessary since these statements were exhaustively repeated in other parts of the main text;
Response: As recommended, both sentences have been removed from the revised manuscript.
Comment 4. Figure 1 should be presented before 2.2 sub-section to improve reading;
Response: We agree with the reviewer and, as suggested, in the revised manuscript, we have moved Figure 1 and its legend to precede Section 2.2.
Comment 5. The concentration of the calculated IC50 could be in micromolar in the tables, as represented in the main text.
Response: Throughout the manuscript and in the main tables (Tables 1, 2 and 3), all ICâ‚…â‚€ values are consistently reported in micromolar units and, as recommended by the reviewer, in the revised version, we have updated all ICâ‚…â‚€ values in the supplementary tables to micromolar for consistency.
Comment 6. The “ratio” mentioned in the main text and in the tables is at really the selective index (SI) present in others chemotherapy studies. Please, consider adjusting to SI your results.
Response: We thank the reviewer for this valuable suggestion. We have adopted the recommended terminology in the revised manuscript and now refer to these ratios as the “Selectivity Index (SI),” as described in Section 2.4 (line 237-245) and Section 3.5 (lines 514-518). Corresponding adjustments have also been made in Tables 1 to 3 (all highlighted in red).